# Cell Dissemination in Pancreatic Cancer

**DOI:** 10.3390/cells11223683

**Published:** 2022-11-19

**Authors:** Jungsun Kim

**Affiliations:** 1Cancer Early Detection Advanced Research Center, Knight Cancer Institute, Oregon Health & Science University, 2720 SW Moody Ave., Portland, OR 97201, USA; kimjungs@ohsu.edu; 2Knight Cancer Institute (Cancer Biology Research Program), Oregon Health & Science University, 2730 S Moody Ave., Portland, OR 97201, USA; 3Department of Molecular and Medical Genetics, Oregon Health & Science University School of Medicine, 3222 SW Research Drive, Portland, OR 97239, USA

**Keywords:** cell dissemination, collective cell migration, epithelial–mesenchymal transition (EMT), adherens junctions

## Abstract

Pancreatic cancer is a disease notorious for its high frequency of recurrence and low survival rate. Surgery is the most effective treatment for localized pancreatic cancer, but most cancer recurs after surgery, and patients die within ten years of diagnosis. The question persists: what makes pancreatic cancer recur and metastasize with such a high frequency? Herein, we review evidence that subclinical dormant pancreatic cancer cells disseminate before developing metastatic or recurring cancer. We then discuss several routes by which pancreatic cancer migrates and the mechanisms by which pancreatic cancer cells adapt. Lastly, we discuss unanswered questions in pancreatic cancer cell migration and our perspectives.

## 1. Introduction

Despite extensive efforts toward developing treatments, the outcome for patients with pancreatic ductal adenocarcinoma (PDAC) remains poor [1]. A leading cause of poor prognosis is frequent recurrence [2,3,4,5]. Surgery is currently the most effective treatment for localized PDAC. However, most patients experience local or distant recurrence after undergoing potentially “curative” surgery, and 95% of these patients die within ten years of diagnosis [2,3,4,5]. The early resectable small T1 stage of PDAC is inferred to progress to the T4 stage in just over one year [6]. Systemic dissemination of dormant PDAC cells occurs before developing into local recurrence or metastases [7,8,9,10]. The dissemination of radiographically undetectable cancer cells currently limits the ability to cure PDAC patients. Why does PDAC have such high cell dissemination, and what types of cell migration exist?

Cell movement (or migration) is essential in embryogenesis, wound healing, host infection, angiogenesis, and carcinogenesis. For example, immune cells migrate through chemotaxis [11,12], neural cells through axon guidance signals and neurotrophins [13,14], and fibroblast cells through the actin-based cytoskeleton [15,16,17,18]. Epithelial cells are not mobile in general. However, several exceptions are observed where stationary epithelial cells can migrate. The first exception is observed during embryogenesis, such as when a primitive streak is formed to initiate the formation of three germ layers (gastrulation) [19]. The second exception is observed during wound healing after tissue injury [20]. Lastly, it is observed in carcinogenesis [21,22]. This review will highlight the epithelial cell movement in pancreatic cancer, which can be a basis for understanding highly recurrent diseases. First, we introduce the evidence for subclinical pancreatic cancer cell dissemination and the organs to which pancreatic cancers metastasize. Next, we discuss the type of cell migrations frequently observed in pancreatic cancer, followed by molecular mechanisms and mediators that lead to such cell dissemination. Lastly, we discuss unanswered questions and potential therapeutic targets.

## 2. Systemic Subclinical Dissemination of Pancreatic Cancer

PDAC is a disease notorious for its high frequency of local and distal metastases. When then, do such metastases occur during disease progression? PDAC is believed to initiate from several preneoplastic lesions with ductal morphology such as pancreatic intraepithelial neoplasia (PanINs), intraductal papillary mucinous neoplasias (IPMNs), and mucinous cystic neoplasias (MCNs) [23]. PanINs are the most common precursor lesion in human and are classified as low grade (PanIN1 and 2) and high grade (PanIN3) based on cytologic atypia severity [23]. Low-grade PanINs are clinically benign, whereas high-grade PanIN3, usually found in pancreata with PDAC, is considered carcinoma in situ [23]. Most PanIN1 lesions (>90%) have activating mutations in the *KRAS* gene [24], and PanIN2 lesions already have genetic alterations in *KRAS*, *CDKN2A*, and *TP53*, which are typically seen in PDAC [24,25].

Murine PDAC models [26,27,28], in which pancreata carry the Kras mutant alleles (*Kras^G12D^*) and/or inactivation of tumor suppressors such as *p53* and *Cdkn2a,* recapitulated disease progression and significantly improved our understanding of PDAC cell migration. In a landmark paper, Rhim et al. showed that pancreatic cancer could be disseminated early in KPCY mice (Pdx1-Cre; Kras^G12D;^ p53^flox/flox^; Rosa-YFP) [10]. They showed a subset of YFP-labeled cells to intermingle with stroma cells, even in the PanINs stages. Further, YFP cell populations were observed in distal organs, such as the liver [10], suggesting that the early disseminating pancreatic cancer cells can be spread out to distal organs without clinical symptoms or forming cancers.

Another murine PDAC study by Pommier et al., also showed that the liver contained dormant, single disseminated cancer cells that had unusual phenotypes with cytokeratin 19 (CK19)-negative and major histocompatibility complex class I (MHC1) [9]. In that paper, to further identify the cell-autonomous “switch” regulating the development of metastatic states in primary tumors, they performed single cell RNAseq on in vitro-sorted E-Cadherin- and E-Cadherin+ cells. They found that unresolved endoplasmic reticulum (ER) stress reduced E-cad and MHC1 expression post-transcriptionally in a subset of primary PDAC. As a result, T cells ended up selecting dormant single disseminated cancer cells by eliminating MHC1+ proliferating cells in the livers of naïve and pre-immunized mice, intrasplenically injected with murine metastatic PDAC [9].

In humans, several studies also indicated the presence of systemic dissemination of pancreatic cancer before clinical diagnosis of metastasis, through computational modeling, exome sequencing, and clinicopathological analysis [7,8,9,29]. Pommier et al. also confirmed the presence of single dormant cancer cells through clinicopathological analysis. They showed that these cells lack expression of CK19 and MHC1 in the livers of patients with PDAC who had no clinically detectable hepatic metastases [9].

Haeno et al. investigated the growth of each cancer and its metastatic probability using two independent databases for a total of 228 PDAC patients: (i) autopsy cohort (*n* = 101) and (ii) adjuvant cohort (*n* = 127) [7]. In the autopsy cohort, data on primary tumor size and metastatic burden were recorded for each patient at diagnosis and autopsy. In addition, approximately half of the patients (*n* = 47) had at least one intermediate time point between diagnosis and autopsy to investigate primary tumor size, local and distant recurrence, and metastases. In the adjuvant cohort, data were recorded for patients who received curative surgeries and adjuvant chemotherapy and radiation therapy. To investigate the growth of each cancer and its metastatic probability, they developed a stochastic exponential mathematical model of PDAC progression and dissemination, with the assumption that metastatic ability is a consequence of a single genetic or epigenetic change and that mutation and dissemination are likely separated in time. Using this stochastic exponential model, they investigated the probability that metastatic cells, as well as cells with the potential to metastasize, are present at the time of diagnosis. They found that not all patients are expected to present with metastatic disease at diagnosis. However, intriguingly, their analysis indicates that all patients can harbor “metastasis-enabled cells” in the primary tumor at the time of diagnosis, even when the primary tumor size is small [7].

Sakamoto et al. performed whole-exome and targeted sequencing of resected primary tumors and matched intrapancreatic recurrences or distant metastases from autopsied tumors from 10 patients with PDAC [8]. Phylogenic studies inferred two distinct evolutionary trajectories by which recurrent disease arose. One group represented the recurrent disease developed from a single residual clonal population (“monophyletic origin”). The other group represented the recurrent diseases seeded by multiple ancestral clones (“polyphyletic origin”). Pairwise Jaccard similarity coefficients for all samples in each patient indicated that monophyletic recurrences were significantly more distant from the primary tumor. In contrast, polyphyletic recurrences were highly related to the primary tumors.

Utilizing mathematical modeling and previously measured metastatic doubling times, they found that the minimum time required to grow from one to a billion cells is 1.82 years. Because clinical metastases occurred much earlier than the required 1.82 years after surgery in patients with the distant disease (6–18 months), at least one of the metastases must have been microscopically seeded before surgery. Their analysis further indicated that recurrent tumors were diverse depending on migration patterns and not phylogenetic hierarchy (e.g., metastasis-to-metastasis seeding or metastasis to local recurrence were also observed [8].

Taken together, these reports suggest the occurrence of systemic subclinical dissemination of pancreatic cancer at the time of diagnosis. Such early dissemination has been observed in other cancers, especially breast cancers [30,31,32].

If so, at what stages can pancreatic cancer migrate in humans? While the murine PDAC model indicated that pancreatic cells at PanIN stages could disseminate into the surrounding tissue and distal organs [10], PanINs are classically considered “carcinoma in situ”. However, a whole-exome sequencing study showed evidence of the movement of PanIN lesions in humans [29]. Makohon-Moore et al. investigated the relationship between PanINs and PDAC by directly comparing somatic nucleotide variants (SNV) and small insertions or deletions (Indels) among laser-captured microdissected (LCM) PDAC and PanINs. In this study, half of all cases (4/8) showed evidence that a common ancestral cell underwent initiation and clonal expansion to form one or more PanINs. Further clonal expansion was driven by additional driver gene mutations in a PanIN cell, eventually leading to the PDAC. Notably, this study selected PanINs from regions anatomically distinct and far removed from that of PDAC. Therefore, this result indicates that a single mutant clone had spread through the pancreatic ductal system to generate co-existing neoplastic lesions [29]. Likewise, targeted sequencing of intraductal papillary mucinous neoplasm (IPMN) and PDAC revealed the intraductal or intraparenchymal spread of noninvasive IPMN cells [33].

In sum, rich evidence in both human and mouse pancreatic cancer suggests that the occurrence of systemic subclinical dissemination of pancreatic cancer at the time of diagnosis can challenge treatment options and early detection strategies.

## 3. PDAC Cell Migration (Figure 1)

The most common metastatic site of PDAC found at autopsy is the liver, followed by the peritoneum, lung, bones, and adrenal glands [34]. PDAC also frequently metastases into lymph nodes in the majority of patients at the time of surgery (67%) [35], and lymph node metastasis determines the stages of PDAC [36,37]. The pathology of metastatic pancreatic cancer is well described elsewhere [34]. So then, what routes does PDAC use to migrate into local or distal organs?

### 3.1. Lymphovascular Invasion 

Mammals have two vascular circulatory systems: lymphatic and blood vasculature [38]. When neoplastic cells invade this vasculature system (e.g., either lymphatic or blood vessels), it is pathologically called “lymphovascular invasion” [39].

#### 3.1.1. Lymphatic Vessel Invasion (LVI)

The lymphatic vessel is a thin tube that carries lymphatic fluid and circulates from tissue to lymph node, from one lymph node to another, or to a larger lymph duct. This lymphatic system plays essential roles in the immune response to inflammation, wound healing, and malignant transformation [38]. Pancreas also has a network of intra- and interlobular lymphatic vessels. The majority of lymphatic vessels in the pancreas are in the interlobular septa of connective tissue, and the intralobular lymphatics are relatively sparse [40]. (Figure 1A) The anatomy and functions of lymphatic systems in the pancreas under physiological conditions are well described in other reviews [40,41].

Lymphatic vessel invasion (LVI) of neoplastic cells is considered a major prerequisite for cancer progression and distant metastasis. Indeed, the presence of LVI correlates to lymph node metastasis and stages in many cancers, including PDAC [36,37]. However, the relative prognostic values of LVI and lymph node metastasis are controversial [42,43]. The LVI of PDAC is well-reviewed elsewhere [42].

#### 3.1.2. Venous Invasion 

The pancreas lies on the posterior abdominal wall, is supplied by multiple branches of the arteries, and is connected to various venous systems [44]. The anatomy of the arterial and venous system in the pancreas is well-reviewed [44]. In this review, I will introduce several essential veins that can be involved in cell dissemination. The venous drainage of the pancreas is to the portal system. In other words, the entire venous drainage of the pancreas eventually reaches the portal vein, which goes to the liver. For example, the portal vein is formed by the confluence of the superior mesenteric, splenic, inferior mesenteric, pancreaticoduodenal, and gastric veins. The tail and body of the pancreas drain into the splenic vein. The head and neck of the pancreas drain into the superior and inferior pancreaticoduodenal veins [44] (Figure 1A).

Microscopic vascular invasion through a small vein (“venous invasion”) occurs in PDAC with a much higher frequency than for any other solid tumors; it is identified in two-thirds of surgically resected PDAC [45,46], or up to 90% of cases when using a more sensitive, three-dimensional detection method (as opposed to 75% in the 2D pancreas in one study) [47]. The neoplastic cells replace endothelial cells and grow along the inner wall of the vessels [45]. (Figure 1B) Intriguingly, more than half of cases that undergo venous invasion mimic well-differentiated PanIN lesions with minimal cytologic and architectural atypia [45]. However, venous-invaded cancers can be distinct from PanINs since they are surrounded by smooth muscle layers [45]. Furthermore, venous invasion is correlated with tumor size, aggressiveness, and other invasive factors (lymphatic invasion, retroperitoneal invasion, serosal invasion), and thus is considered a significant factor for a poor prognosis [45,46]. The venous cells invaded by PDAC are expected to drain directly into the portal vein, resulting in liver metastasis [48,49,50]. Indeed, a significant correlation was found between microscopic venous invasion and liver metastasis [46], and venous invasion is considered one crucial factor that makes PDAC so deadly [50].

### 3.2. Intraductal Migration (Figure 1C)

“Intraductal carcinoma foci” [51] or “cancerization of ducts (COD)” [33] is a migration pattern unique to PDAC. With this migration pattern, cancer cells can infiltrate back into the pancreatic duct and spread intraductally along the pancreatic ductal system [33,51,52]. COD is widely observed among 3–90% of PDAC cases [33,51,52]. Recent high-resolution 3D images of human pancreata confirm that COD is present with the bridging of neoplastic cells across the duct lumen [53]. Intraductal spread of carcinoma is frequent in well-differentiated PDAC [51]. While distinguishing COD and high-grade PanINs is difficult, since both are intraductal epithelial lesions with similar histological appearances, COD is generally suspected when it is immediately adjacent to invasive carcinoma. Further differentiation between suspected COD and PanINs can be made with mutational profiling of TP53 and SMAD4 genes, which are late events in the genetic progression of PDAC and rare in PanINs [33].

Elucidating the mechanisms of the intraductal spread of PDAC is important since the pancreatic ductal tree can be another route of PDAC dissemination. Moreover, intraductal spread hampers the designation of the appropriate surgical resection. So then, when does PDAC initiate spreading intraductally, and how does it happen? It is difficult to define whether carcinoma initially spreads throughout ducts without invasion or infiltrating carcinoma components secondarily involving the ducts. However, sequencing studies indicated that neoplastic cells could spread intraductally even in the noninvasive stage; A study with whole-exome sequencing of PanINs suggests tiny clusters of early neoplastic lesions budding off into the lumen [29]. Likewise, targeted sequencing of IPMN and PDAC in the remnant pancreas after IPMN resection revealed an intraductal or intraparenchymal spread of noninvasive IPMN cells [33].

Although the mechanisms by which PDAC cells spread along the ductal system remain to be defined, a study allows us to infer the potential factors for differing migration patterns of the ductal cells [54]. Messal et al. investigated how cellular tension can contribute to the changes in tissue architecture and tumor formation during cancer initiation in the normal pancreatic duct using a three-dimensional whole-organ imaging technique [54]. They triggered epithelial deformation by inducing conditional activation of *Kras^G12D^* alleles with concomitant deletion of either *p53* or *Fbw7* tumor suppressors. Two morphologically distinct lesions were induced in pancreatic ducts transformed by *KRAS^G12D^* activation (exophytic vs. endophytic); exophytic lesions evaginated basally away from the ductal lumen, thus extending the duct lumen and forming globular structures. In contrast, endophytic lesions invaginated apically towards the ductal lumen, grew into the ductal lumen in a papillary manner, and progressed to intratubular neoplasia with local obstruction of the duct lumen. Thus, while exophytic lesions showed traditional disseminating lesions through the extracellular matrix, endophytic lesions were very similar to the intraductal migrating tumors, which grow into the ductal lumen. Likewise, mouse and human exophytic lesions recruited cancer-associated fibroblasts more efficiently than the endophytic lesions that grow into the duct lumen [54].

Tissue morphology is determined by cellular tension and contractility, driven by the actomyosin cortex [54]. Thus, to further define the potential mechanisms by which these two distinct invasions occurred, they quantified the distribution of the key cortex components F-actin and phosphorylated myosin light chain 2 (pMLC2) in single cells in the KPC mice before neoplastic lesions occurred. While the intensity of F-actin and pMLC2 staining was higher apically than basally in wild-type cells, this apical–basal gradient was strongly reduced upon transformation. Phosphatase activity was essential for pMLC2 apical–basal polarization since phosphatase inhibition disrupted apical–basal pMLC2 gradients in wild-type organoids. They further showed that the KRAS-MEK-ERK pathway could be a driver of pMLC2 gradient shift since apical pMLC2 accumulation was partially restored upon MEK inhibition in KPC organoids [54].

They assumed that apical and basal surface tension was proportional to the relative fluorescence intensities of pMLC2. Thus, to test whether the pMLC2 perturbation in transformed cells could explain lesion deformation, they developed a computational model of the pancreatic duct using a “3D vortex model simulation that integrates ductal cell geometry, the tensions of apical, basal, and lateral surface, and cell volume conservation [54]. This model introduced a single cell with surface tensions based on experimentally measured average pMLC2 fluorescence intensity changes. They also simulated the successive rounds of cell division based on experimentally observed cell counts at different time points. This computational modeling revealed that the size of ductal tissues determines the transition from exophytic (smaller duct) to endophytic lesions (larger duct > 17 um diameter) and that hyperproliferation alone is insufficient to trigger diameter-dependence of exophytic and endophytic lesions, as the endophytic lesion growth requires altered apical–basal tension [54].

In sum, this study offers one potential trigger of intraductal migration. Further investigation will be required to understand whether the size of ducts can pre-determine specific migration patterns, routes, and severity to disseminate into other organs in vivo.

### 3.3. Neural Invasion (Figure 1C)

#### 3.3.1. Rich Neuronal Environment of the Pancreas

The pancreas has abundant nerve supplies, mainly from the celiac plexus, mesenteric arterial plexus, and splanchnic plexus [55,56]. It is innervated and regulated by the autonomic nervous system with sensory, parasympathetic, and sympathetic neurons [57]. The roles of nerve systems in PDAC have been examined in murine PDAC models. For example, KPC mice, deficient in the cholinergic receptor muscarinic1 for the parasympathetic nervous system, have accelerated PDAC progression [58], whereas KPC mice ablated for neonatal sensory fibers have slower PanIN progression [59]. The dopamine receptor D2 is upregulated in PDAC, and inhibition of the receptor reduces PDAC proliferation and migration in mice [60]. Altogether, all autonomic neurons play a role in PDAC pathogenesis in mice.

#### 3.3.2. Perineural Invasion

PDAC cells commonly encroach along nerves (“perineural invasion”) at a prevalence higher than other cancer types [55,56,61,62,63,64]. Perineural invasion is observed among 70–98% of PDAC cases, and 75% of stage I PDAC cases already have perineural invasion [55,56,61,62,63,64]. Perineural invasion is associated with poor prognosis and peritoneal metastasis [55,56,61,62,63,64,65], the second most common metastatic site of PDAC. Perineural invasion can occur without lymphatic or vascular invasion and does not correlate with the tumor size [56]. In addition, communications between the pancreas and neurons occur before PDAC establishment and infiltrating neural cells (“innervation”) increase in the pancreas at precursor stages [59,66,67]. These reports suggest that perineural invasion may represent an initial step of metastasis and contribute to the early dissemination of PDAC.

However, neuronal types involved in the perineural invasion of PDAC remain unclear. Extensive reports on the perineural invasion of PDAC are mainly based on pathological examination of surgically resected tumors [55,56,66]. Ex vivo co-cultures have been used to model perineural invasion or innervation of human PDAC. However, these studies use dorsal root-ganglion (DRG) [67,68], which includes glial cells, a few neural stem cells, and sensory neurons; therefore, it is a caveat that direct interaction between pancreatic cancer and neuronal cells cannot be assumed. Further careful investigation will be required to identify the types of nerve cells involved in perineural invasion.

#### 3.3.3. Axon Guidance Signals

Several factors, such as chemokine, neurotrophic factors, and axon guidance signals, are involved in the perineural invasion of cancer. Many studies have focused on chemokines and neurotrophic factors in the perineural invasion of PDAC and have been reviewed elsewhere [69,70,71,72]. Herein, we will discuss axon guidance signals in the perineural invasion.

Elucidating how cancer cells accurately recognize and attract (or repulse) neural cells to intrude into target sites is essential for understanding cell dissemination and suggesting targets to reduce recurrence in PDAC. Indeed, the class of genes traditionally described for their roles in axon guidance is recently implicated in neural invasion [73] as well as cancer cell growth, invasion, and angiogenesis in multiple cancers [74,75,76]. The axon guidance molecules are subdivided into attractant and repulsive cues and cell migration direction [76,77]. However, axon guidance signaling has received relatively little attention in the perineural invasion of PDAC. A study of a cohort of patients (*n* = 142) with early (stages 1 and 2) PDAC revealed recurrent and clinically impactful aberrations in axon guidance pathway genes [78]. Specifically, the study demonstrates that 5% of cases have *SLIT2* or *ROBO2* mutations with focal copy-number losses and that 18% harbor genomic amplification of the class 3 Semaphorin genes *SEMA3A* and *SEMA3E*. The study further shows that poor patient survival is associated with low ROBO2 mRNA expression and high mRNA levels of SEMA3A and its receptor PLNXNA1. Thus, this study shows dysregulation of axon guidance signaling genes in stages 1 and 2 PDAC.

Two studies have shown axon guidance molecule functions in perineural invasion or metastasis [73,79]. Jurcak et al. show that the attractant guidance cues SEMA3D/PLXND1 axis promotes tumor metastasis using murine, a PDAC/DRG co-culture system, and an orthotopic model with a sensory neuron-specific knockout of *Plxnd1* [73]. In contrast, Gohrig et al. show that SLIT2/ROBO inhibits perineural invasion and metastasis of PDAC using the PDAC/DRG co-culture system and orthotopic model [79]. However, these studies mainly focused on the impacts of axon guidance signals in malignant PDAC and DRG. Thus, it remains to be determined what and how cancer cell-master factors drive the dysregulation of axon guidance signals and how axon guidance signals regulate the interaction of normal and early neoplastic cells with neural cells, subsequently initiating metastases.

In sum, PDAC cells can disseminate through multiple routes, such as venous invasion, intraductal invasion, and neural invasion. How, then, do PDAC cells migrate?

## 4. Mechanisms behind the Migration of Pancreatic Cancer Cells

Both intrinsic and extrinsic factors impact PDAC cell dissemination. PDAC has a unique microenvironment, including highly infiltrating cancer-associated fibroblasts and an immune-suppressive microenvironment. Reviews of the pancreatic cancer microenvironment have been extensively discussed elsewhere [80,81,82]. This section will focus on cell-intrinsic components that can mediate cell dissemination.

### 4.1. Polyclonal Origin or Monoclonal Origin

There have been several genetic studies with phylogenetic analyses of the heterogeneous origins of PDAC in humans. For example, regional whole-exome sequencing of paired precursors and PDAC showed heterogeneous precursor stages [29,83]. Such heterogenicity is also observed in recurrent diseases [8]. Whole exome sequencing of matched resected primary tumors and intrapancreatic recurrences or distant metastases showed that recurrent diseases could be seeded by multiple ancestral clones (“polyphyletic origin”) as well as a single residual clonal population (“monophyletic origin”). However, intriguingly, while polyphyletic recurrence was related to the primary tumors, monophyletic recurrence was significantly more distant from primary tumors [8], implying that selective pressure could reduce clonal diversity during PDAC progression.

In mice, Maddipati and Stanger investigated how such heterogeneity occurs and evolves during PDAC using the confetti-lineage system [84,85]. Cre-mediated recombination leads to stochastic expression of one of four fluorescent proteins in any given cell (Pdx1-Cre^ERTM^; Kras^G12D^; p53^wt/flox^; Rosa^Confett^ mice, KPCX mice) [85]. First, they investigated precursor lesion clonality by examining serial pancreatic sections from PanIN and ADM stages (8-to-10-week-old KPCX mice). While approximately 24% of all ADM were polychromatic clones, most PanINs (97%) examined were monochromatic clones. These data indicate that clonal diversity is reduced during premalignant progression [85].

They further demonstrated that metastases often involved seeding by more than one clone and the subsequent cellular outgrowth depends on the metastatic sites [85]. Specifically, they investigated the clonality of metastatic tumors by examining tumors in the peritoneal wall, diaphragm, liver, and lung. Interestingly, while nearly 80% of peritoneal and diaphragm metastatic tumors were polychromatic, extensive polyclonal metastases were not observed in liver and lung metastases; Most liver and lung metastatic tumors were single or nano metastases (2–10 cells), with only 10% to 26% having more than ten cells. They found that bichromatic cellular aggregates or clusters of fluorescent tumor cells were presented in the ascites fluid or blood from tumor-bearing KPCX mice. Thus, they confirmed that polyclonal metastases are more likely derived from the outgrowth of polyclonal lesions at the seeding. Further in vivo mixing experiments (injecting single cells or aggregated cells) indicated that, while polychromatic lesions constitute the majority (70%) of metastases following injection of cell clusters, only monochromatic lesions were seen following injection of the single-cell suspension [85].

In sum, human and mouse studies demonstrate that, while PDAC occurs from a heterogenous origin, the clonal diversity decreases during disease progression.

### 4.2. Single-Cell Migration vs. Collective Cell Migration

Maddipati et al. suggested that cell clusters are more efficient than single cells at forming metastasis, regardless of metastatic sites [85]. Consistently, recent emerging studies have demonstrated that epithelial cancer cells can migrate as “cell clusters” in various tumor types [86,87,88,89,90,91] (often called “collective cell migration”) as opposed to “single-cell migration”. Why, then, are cell clusters shown to be more efficient?

Normal epithelial cells without mesenchymal phenotypes or transformation with oncogenes are programmed to die if they are exfoliated as single cells (the type of apoptosis, which is induced by disrupting the interaction of epithelial cells with matrix interaction, is called “anoikis”) [92]. Moreover, single circulating epithelial cells in the blood can be easily tracked and die out by immune surveillance [88,89]. Importantly, disseminated cancer cell clusters show enhanced survival and metastatic phenotypes in murine breast and PDAC models with multicolor lineage tracing [85,93], and clustered circulating tumor cells in the blood are associated with poorer patient prognosis in various cancer types [88,89]. Moreover, disseminating cancer cell clusters are implicated in early cancer progression [94]. The 3D imaging analysis of clinical tissues shows that venous invading cancers have intact epithelial cell clusters, suggesting clustered cell migration in patients with PDAC [47].

In sum, while PDAC has been known to disseminate as a single cell in general, recent studies demonstrated that PDAC cells could migrate as cell clusters (“collective cell migration”). So, what makes pancreata determine their migration patterns?

### 4.3. Cell Adherens Junctions and Epithelial–Mesenchymal Transition (EMT)

The integrity and identity of epithelial cells are maintained by specific cell–cell interactions through adhesion complexes such as adherens junctions, tight junctions, and desmosomes. The loss of apical–basal polarity can lead to the destabilization of adhesion complexes. E-cadherin, the main component of the adherens junctions [95,96], is stabilized with catenin proteins such as beta-catenin and p120 at the adherens junctions [97,98,99,100]. Loss of E-cadherin triggers dissembling cell adherens junctions, and it subsequently renders stationary epithelial cells able to gain a migratory phenotype called “epithelial–mesenchymal transition (EMT)” [101,102] As a result, epithelial cells loosen their cell–cell junctions to become motile. They can degrade the basement membrane, be squeezed out of the extracellular matrix, migrate, and invade nearby stroma cells. Detached tumors can penetrate the vascular or lymphatic vessels to enter the bloodstream for systemic circulation (intravasation). Therefore, EMT is considered the most critical factor in cancer metastasis. The detailed criteria to define EMT and its mechanisms are well-discussed elsewhere [101,102].

Growing evidence shows that such an EMT process is not an “on-and-off” process. Instead, EMT is observed as a continuum in many cases. In other words, the EMT process is often incomplete, resulting in cells displaying both epithelial and mesenchymal characteristics [101] (called “partial EMT [103]” or epithelial–mesenchymal plasticity: EMP [102]), and is frequently seen in cancer, embryogenesis, and wound healing process.

Such EMP status is also observed in the murine PDAC model. Aiello et al. showed that some cancer cells express both epithelial and mesenchymal genes but lose epithelial phenotypes (called “partial EMT”), as opposed to complete loss of E-cadherin mRNA (called “complete EMT”), in a lineage-labeled PDAC murine model [103]. Cancer cells that undergo partial EMT are associated with collective cell migration, as seen in many tumor types [87,88,90,91,94,104]. Mechanistically, Aiello et al. further showed that internalization of the intracellular domain of shed E-cadherin allows cells to have both mesenchymal and epithelial gene signatures. Therefore, that paper provides evidence of “partial EMT” collective cell migration and the relationship between EMT and the invasive/cell migration phenotype in vivo in PDAC [103]. They further showed that prolonged calcium signaling is sufficient to induce partial EMT, characterized by internalized E-cadherin and co-expression of epithelial and mesenchymal markers, as well as intraductal cellular migration and invasion [105]. However, what master regulators induce partial EMT upon calcium influx remains unknown.

## 5. Questions for Future Investigation

Advances in the murine model, imaging technologies, and genomic studies have significantly improved our knowledge of the highly metastatic PDAC. We have learned about several migration processes that PDAC harnesses and the potential mechanisms involved in these migrations. The eventual goal of these combined works would be the development of new therapeutic targets. It is crucial to understand pancreatic cancer as a systemic disease and reconstruct potential regulatory networks to uncover master factors that may govern PDAC cell dissemination and potentially serve as these therapeutic targets.

### 5.1. What Master Factors Contribute to Partial EMT and Collective Cell Migration?

Deciphering the mechanisms by which pancreatic cancer cells detach from tumor mass during cancer progression is critical in understanding PDAC dissemination. Moreover, when cells respond to extrinsic cues, cells use adherens junctions along their path to maintain the appropriate trajectory [106], thereby emphasizing the importance of cell adhesion complexes in targeting cancer cells throughout cell migration. However, the mechanisms by which PDAC intrinsically initiates and coordinates dissemination in cell clusters and how cancer cells regulate adherens junctions to interact with the neural, venous, and ductal system throughout PDAC progression are currently unclear.

While partial EMT has been well-described in the murine PDAC model [103], it is unclear to what extent partial EMT or conventional EMT contributes to PDAC development and what types of invasions function in humans. For example, investigating clinical samples using a newly developed 3D imaging system indirectly demonstrated that “cancerization of duct” or “venous invasion” may not adopt conventional EMT since they mimic high-grade PanINs or well-differentiated PDAC [47,53]. Likewise, collective cell migration has not been well-studied in human pancreatic cancer. Therefore, further mechanistic studies using human model systems will be required to evaluate whether partial EMT and collective cell migrations occur and to what extent partial EMT and collective cell migrations impact human PDAC progression.

Moreover, it is not fully elucidated what master factors in a cancer cell drive the partial loss of E-cadherin as a route for EMT in PDAC progression and whether dysregulation of these factors can prevent or promote cancer cells from disseminating as cell clusters and metastases. Prolonged calcium signaling triggers partial EMT. An increase in intracellular Ca^2+^ results in the activation of several downstream mediators. However, it remains unclear what cell intrinsic mediators trigger partial EMT. Further studies will be required to investigate to what extent cancer intrinsic and extrinsic factors contribute to this partial EMT and further cell dissemination.

### 5.2. Understanding the Relationship between Metastatic Sites and Various Invasion Routes

As described above, pancreatic cancer has been shown to adopt various routes to migrate into distal organs, including perineural invasion, intraductal invasion, and venous invasions. These invasion routes have been individually identified from clinical studies using specimens. The questions remain whether these invasion paths occur independently or intermingled with each other and whether specific invasion routes can pre-determine metastatic sites to which PDAC migrates. For example, perineural and venous invasions are associated with peritoneum and liver metastases, respectively [48,49,50,65]. However, mechanisms by which perineural invasion or venous invasion leads to peritoneal metastasis or liver metastasis remain undefined. Mainly, venous invasion and intraductal invasion have not been well-investigated in a model system. What are the drivers that trigger venous invasion? Likewise, it is unclear whether venous invasion is also linked to peritoneum migration and/or perineural invasion. However, Bandyopadhyay et al. reported that PDAC existed as “isolated solitary ducts” in the section of the retroperitoneal resection margin and peripancreatic soft tissue. Further, a small subset of these units represents vascular invasion [107], indicating that venous invasion can also participate in peritoneum invasion.

It is unclear whether and to what extent perineural invasion, venous invasion, and intraductal migrations are involved in PDAC cell dissemination. Elucidating roles of perineural invasion, intraductal migration, and venous invasion in cell dissemination and, subsequently, local and distant recurrence is vital to understanding PDAC and developing effective treatment strategies.

### 5.3. Organotropism of Pancreatic Cancer

PDAC predominantly metastasizes into the liver, peritoneum, and lung. Pre-metastatic environments are known to be established in the liver, and extensive efforts are made to identify factors that determine organotropisms in the liver. For example, exosomes in primary PDAC result in pre-metastatic niches in the liver [10]. Unresolved ER stress in primary PDAC determines a dormant characteristic of disseminated cells in the liver [9]. However, little attention is given to autonomous pancreatic factors mediating organotropisms, except for a study that showed that PDAC p120 catenin-mediated epithelial identity could determine liver and lung metastatic organotropism [100]. They showed that epithelial integrity through p120 catenin/E-cadherin was required for liver metastasis in human and murine PDAC. In contrast, loss of p120 in both alleles mediated lung metastasis [100].

While extensive studies show evidence that the liver can provide a niche for disseminating subclinical dormant PDAC, peritoneum metastasis is relatively less studied, despite its significant contribution to PDAC. Fifty percent of patients with PDAC show metastasis in the peritoneum at the time of death, making this the second most common site after the liver [34]. Approximately 9% of PDAC cases already have established peritoneum metastases at the time of diagnosis [108,109]. Peritoneum metastasis is considered a local metastasis, and it is still unclear whether local peritoneum recurrence results in liver metastasis or if peritoneum metastasis is a distinct process.

Most peritoneal metastatic tumors were polychromatic tumors, while liver or lung metastatic tumors rarely contained extensive polyclonal metastases in the lineage-tracing PDAC murine model [85]. Exome sequencing of recurrent tumors with their matched primary tumors indicated that recurrent tumors are less likely to be phylogenetic events [8]. It may suggest that peritoneal microenvironments are more susceptible to seeding cancers.

In contrast, lymph node metastasis plays an important role in determining the stages of PDAC [35,36]. However, clinical studies indicate that long-term survivors of PDAC have lymph node metastases [50]. Indeed, there are similar survival rates among PDAC patients who have and who do not have lymph node metastasis [110]. Therefore, further studies will be required to better understand PDAC cell fates and prognoses depending on various types.

How does the PDAC cell migrate into the distal organ in the early stages? While genomic studies demonstrated the evidence of intraductal migration of PanINs or IPMN within the pancreata, the consequence of such migration is unclear. Therefore, further investigation is required to understand whether such migrated precursors can survive in the lumen of the pancreatic duct and reach the distal organ.

Lastly, what is the outcome of such early dissemination before showing metastatic disease in PDAC and other cancers? Breast cancers frequently show such early dissemination [30,31,32], and the risk of early-stage breast cancer recurrence after treatment is relatively high [111]. However, the average survival rate for breast cancer patients is excellent owing to improved early detection technologies and targeted treatment options. That being said, we cannot ignore the biological differences between PDAC and breast cancer. For example, LVI or lymph node metastasis is an essential route to metastasize into other organs and a crucial parameter to determine the prognosis of breast cancers. However, the relative prognostic values of LVI and lymph node metastasis are controversial in PDAC [42,43]. Moreover, PDAC has various unique routes to other organs, such as perineural, venous, and intraductal invasions [50]. Therefore, further studies will be required to understand the consequence of each dissemination and master factors to trigger.

## 6. Closing Remarks

Over the past decades, spatial genomics and imaging technology have identified significant pathophysiological events involved in PDAC cell dissemination. Integrating this information into the appropriate model system and elucidating the mechanisms behind the pathophysiological process would help dissect and understand neoplastic diseases and further identify therapeutic candidates that could intervene in PDAC cell dissemination and progression.

**Figure 1 cells-11-03683-f001:**
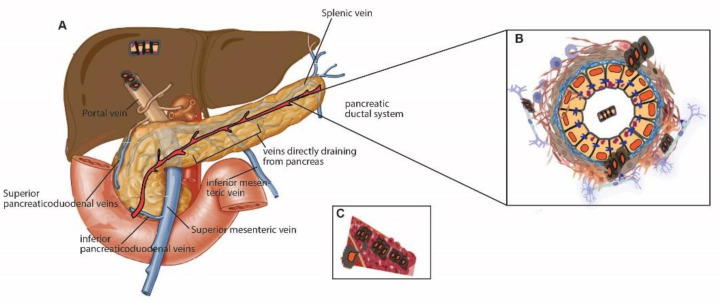
Cancer cell migration in the pancreas. (**A**) Topographical view of venous drainage of the pancreas. The portal vein is formed by the confluence of the superior mesenteric, splenic, inferior mesenteric, pancreaticoduodenal, and gastric veins. The tail and body of the pancreas drain into the splenic vein. The head and neck of the pancreas drain into the superior and inferior pancreaticoduodenal veins [44]. (**B**,**C**) Depiction of pancreatic ductal cells. Exfoliated cells invade the extracellular matrix by breaking down basement membrane proteins. Migration of exfoliated cells through the ductal lumen (L) is not common but is occasionally seen in pancreatic cancer. The invading PDAC cells can further interact with nerve (**B**) or venous systems (**C**) in the pancreata. The use of anatomical diagrams of the pancreas and veins were granted from Elsevier (copyright) [44] and modified using Adobe illustrator and BioRender.

## Data Availability

Not applicable.

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
