# Peer review of "Cell Dissemination in Pancreatic Cancer"

_cells, 2022, doi:10.3390/cells11223683_

Round 1

Reviewer 1 Report

The article summarizes some of the major research findings regarding molecular mechanisms and modeling of metastatic spread of pancreatic adenocarcinoma. The paper contains multiple scattered minor grammatical mistakes or awkward wording that should be corrected or clarified. For example:

"As a result, T cells ended up selecting dormant single single disseminated cancer cells by eliminating MHC1+ proliferating cells..."

The author otherwise nicely summarizes the key pathways involved in metastatic spread including current limitations in knowledge about this highly lethal malignancy. I recommend manuscript acceptance after grammatical correction of the manuscript.

Author Response

Thank you for your constructive feedback on my manuscript. Two professional English-speaking scientific writers edited and proofread my revised manuscript.

Reviewer 2 Report

In this review article, the author comprehensively described recent findings of PDAC cell dissemination with a number of references. First, the author highlighted systemic dissemination of pre-malignant pancreatic cells that has been confirmed in two important studies of mouse models. Comprehensive transcriptomic analysis and mathematical modeling in human were also described, which indicated systemic subclinical dissemination of pancreatic cancer cells at the time of diagnosis and heterogeneous cancer cell expansion patterns from a single origin and multiple clones. The author next classified the disseminating routes of pancreatic cancer cells, venous invasion, intraductal migration, neural invasion. Then, the mechanisms behind the dissemination were described, including the relation of monophyletic or polyphyletic origin and the metastatic sites, single cell or collective cell migration, and EMT. Finally, the author raised questions that remain to be investigated, the master regulator of partial EMT and collective cell migration, the relation of metastatic sites and dissemination routes, metastatic niche in each metastatic site with the prognosis, and the mechanisms of early stage migration. This article seems well organized and reads well. I just raised a couple of points to be addressed.

1) The author described a possibility of occurrence of systemic subclinical dissemination of premalignant cells. This is a very interesting and important phenomenon. Is such a phenomenon specific for pancreatic cancer and is it one of the reasons of the poor prognosis of pancreatic cancer? Is there any similar report in other types of cancer? Can the author mention about it?

2) The author described several routes of dissemination of pancreatic cancer, but there was no mention about lymphatic vessel invasion, which seems a lack for the comprehensive review.

3) If the author can provide a graphic image of this review, it will help readers a lot.

Minor points

4) Section 4 title looks odd in Page 6.

5) In line 1 of Page 10, lung metastasis is not rare in pancreatic cancer.

6) Ref.1, the cancer statistics, had better be updated to the newest one. 

Author Response

Thank you for your constructive feedback on my manuscript

Answers to question #1. I described the early dissemination that occurred in other cancers on page 3 (lines 120-121) and the differences between pancreatic and breast cancers on page 11 (lines 516-528).

Answers to question #2. I incorporated lymphovascular invasion as section2.1 and introduced lymphatic vessel invasion along with the venous invasion on page 4 (lines 148-176)

Answers to question #3. I incorporated a graphical diagram for cell dissemination (figure 1).

Answers to Minor point #4. I corrected the format.

Answers to Minor point #5. I removed the “rare” word.

Answers to Minor point #6) I replaced it with the latest updated reference.